# Optimization of Antibacterial, Physical and Mechanical Properties of Novel Chitosan/Olibanum Gum Film for Food Packaging Application

**DOI:** 10.3390/polym14193960

**Published:** 2022-09-22

**Authors:** Maedeh Salavati Hamedani, Mohammadreza Rezaeigolestani, Mohammad Mohsenzadeh

**Affiliations:** Department of Food Hygiene and Aquaculture, Faculty of Veterinary Medicine, Ferdowsi University of Mashhad (FUM), Mashhad 9177948974, Iran

**Keywords:** chitosan, olibanum gum, food packaging, response surface methodology

## Abstract

Chitosan-based films are promising active biodegradable materials with the ability to be enhanced by different materials, including gums. This study aims to optimize the physical (transmittance, water vapor permeability and water solubility), mechanical (tensile strength and elongation at break) and antibacterial (against *Staphylococcus aureus* and *Salmonella* Typhimurium) properties of newly fabricated chitosan/olibanum gum (CH/OG) films as a function of different levels of CH (0.5, 0.75, 1, 1.25 and 1.5% *w*/*v*) and OG (0.125, 0.25, 0.375, 0.5 and 0.625% *w*/*v*), using response surface methodology (RSM). Moreover, Fourier-transform infrared spectroscopy (FTIR), scanning electron microscope (SEM) and differential scanning calorimetry (DSC) were used to better characterize the fabricated films. RSM analysis results showed the significant fitting of all dependent variable responses to the quadratic polynomial model. To attain the desirable physical, mechanical and antibacterial responses, the optimal concentrations were 1.31% (*w*/*v*) CH and 0.3% (*w*/*v*) OG. The encouraging antibacterial, physical and mechanical properties of the developed composites support the application of chitosan/gum blends in active food packaging, particularly for perishable foodstuffs, such as meat and horticultural products.

## 1. Introduction

Many researchers from different disciplines are focused on the development of sustainable bio-based materials that can be used in various sectors, including food packaging, for partially replacing petroleum-based plastics. Business Communications Company (BCC) research anticipates that the global bioplastics market size is expected to grow from 1.6 million metric tons in 2018 to 2.7 million metric tons by 2023, at a compound annual growth rate of 11.7% [1,2]. In fact, an important part of this growth is in response to the growing consumer demand for using eco-friendly materials, especially in food packaging. Food packaging materials based on biopolymers are an attractive, eco-friendly alternative to synthetic petroleum-based plastics, and they can extend the shelf life of food products via various mechanisms, including reducing gas, water vapor and flavor transmission, and sometimes releasing active compounds into the food matrix [3].

Among different types of biopolymers, chitosan (CH) gained attention due to its valuable physical, mechanical and biological properties. CH is the deacetylated product of chitin, the second most abundant polysaccharide in nature, and is industrially produced in different quality grades [4]. This cationic non-toxic biopolymer has a good film-forming ability and, owing to its antimicrobial activity CH-based films, can be classified in the active film category [5,6]. However, there are some factors which limit the applicability of CH in food packaging, such as its poor mechanical and water vapor barrier properties and cost, compared to synthetic petroleum-based packaging materials. Various strategies, including the incorporation of plasticizer, chemical crosslinking and polymer blending, were followed by researchers to overcome these shortcomings [7]. Among those approaches, polymer blending is a cost-effective and simple method that can bring some new attractive features to the base polymer by adding other materials [8].

Different types of materials, including gums, carbohydrates, active agents and nanoparticles, are blended with chitosan or other film-forming polymers to improve the functional properties of the composites [9,10]. CH/gum blends are promising forms of binary blend films, aimed at improving the overall applicability of the resultant composites, including food packaging applications. Various gums, such as arabic [11,12], tara [1] and xanthan gums [13] with varying physico-chemical properties were successfully combined with CH. In fact, gums via different mechanisms, including electrostatic interactions, emulsifying action, retention of active agents and acting as antimicrobial or antioxidant, could efficiently modify the resulting composites [11,12,14].

Among the well-known natural gums, olibanum gum (OG) has a long historical application in medicine and foods [15]. Olibanum is an aromatic gum resin obtained from trees of the Boswellia genus, such as *Boswellia serrata*. This genus belongs to the Burseraceae family and mainly grows in India, East Africa and Arabia [16]. OG mainly consists of acid resin, gum and essential oil. The gum portion contains various water-soluble polysaccharides, while resin encompasses mainly a resin acid: namely, boswellic acid. The film-forming ability of the OG resin is well demonstrated [17]. Additionally, it is shown that boswellic acid can enhance some physical and mechanical properties of CH/poly (vinyl alcohol) [18]. It should also be mentioned that the essential oil of OG, through its active constituents, possess antimicrobial effects against food-borne pathogens [19]. The advantages of this gum over synthetic polymers can be low cost and toxicity, abundant availability, biodegradability and biocompatibility [17]. Combined together, OG has an attractive potential to be used in food packaging, particularly in combination with other bio-based packaging materials.

To date, mainly the essential oil of OG is studied as an active agent in packaging materials, and negligible efforts have been made to benefit the capability of OG in the development of new polymers with superior physico-mechanical properties.

The aim of the present research is to evaluate and optimize the physical, mechanical and antibacterial properties of CH films incorporated with OG for their potential application in food packaging. Response surface methodology (RSM) was applied as an efficient statistical method for simultaneously examining and optimizing the response variables.

## 2. Materials and Methods

### 2.1. Materials and Preparation of Chitosan Films

Medium molecular weight chitosan (100 kDa) with a deacetylation degree of 75%, as specified by supplier, acetic acid, glycerol and all culture media and diluents used for microbiological tests were obtained from Sigma-Aldrich (Milwaukee, WI, USA). Olibanum gum was purchased from Johare Taem Co. (Mashhad, Iran), and all chemical reagents used in this study were of analytical grade.

The chitosan film-forming solutions (FFS) were prepared according to a method described by Xu et al., 2019 with some modifications [12]. Certain amounts of chitosan powder (Table 1) were dispersed in the aqueous solution of acetic acid (1%, *v*/*v*), and glycerol as a plasticizer, at a concentration of 0.75 g glycerol/g chitosan, was added to the solution. The resultant mixture was then stirred at 60 °C by a heater–stirrer for 3 h.

In parallel, to prepare the OG solution, the method of Choi et al. was used [20]. In brief, the gum was powdered in a high-speed mechanical blender (Model 1000, Asan Toos Shargh, Tehran, Iran) to obtain a smooth powder. Then, OG solutions were prepared in distilled water by adding certain amounts of OG (Table 1) and stirring overnight at ambient temperature. The resultant mixture was centrifuged at 1500× *g* for 10 min. Subsequently, the supernatant was again centrifuged at 2500× *g* for 10 min and successively at 10,000× *g* for 20 min to remove insoluble materials.

The FFSs were prepared by gradually adding OG solutions into CH solutions. Then, the obtained FFSs were vigorously homogenized at 12,000 rpm for 4 min using a homogenizer (IKA T18-digital ultra turrax, Staufen, Germany). Deaeration of FFSs was carried out by an ultrasound device (300 UPS, FAPAN, Tehran, Iran). The degassed solutions were cast on glass Petri dishes (10 cm diameter) inside an oven at 37 °C for 48 h. Finally, the dried composites were placed in polyethylene plastic until tested.

### 2.2. Characterization of the Films

#### 2.2.1. Transmittance

The visible light barrier performance of chitosan films was measured using a UV-VIS spectrophotometer (Optizen 2120UV plus, Mecasys Co. Ltd., Daejeon, Korea). Each film sample was cut into a strip of 3 × 1 cm and placed directly in a spectrophotometer cell. Transmittance was measured at a fixed wavelength of 660 nm [21]. The mean of three measurements was recorded for each film specimen.

#### 2.2.2. Water Vapor Permeability (WVP)

For WVP determination (g/m h Pa) of the composites, the standard method ASTM E96–95 was used [22]. First, film specimens were carefully placed and sealed on top of permeability cups containing CaCl_2_ (0% relative humidity). Then, the cups were weighed and placed in a desiccator at 25 °C with a relative humidity of 75%. A saturated NaCl solution was used to maintain the humidity. The cups were weighed every 2 h for 18 h. The WVP was calculated by the following equation (Equation (1)):WVP = Δm X/A Δt Δp(1)
where Δm/Δt is the weight loss or gain of cups as a function of time (g/h), X is the thickness of the film (m), A is the test area (m^2^) and Δp is the partial pressure difference across the film (Pa).

#### 2.2.3. Thermal Properties

The thermal properties of the films were determined using a differential scanning calorimeter (DSC, 60 model, Shimadzu, Kyoto, Japan) [23]. Approximately 5 mg of the films was cut and hermetically sealed in an aluminum pan. The film samples were heated at a rate of 10 °C/min from 25 to 250 °C. An empty sealed aluminum pan was used as a reference.

#### 2.2.4. Water Solubility

Films solubility was measured in accordance with the methodology described by Gontard et al., 1994 [24]. Approximately 2 cm in diameter of the films were cut and weighed before and after the oven drying process at 105 °C for 24 h. Then, the samples were immersed in 50 mL of distilled water at 25 ± 1 °C and subjected to constant stirring at 125 rpm for 24 h. Afterward, in order to determine the dry matter content of the solubilized films, the CH films were dried and weighed as previously explained. The water solubility of the films was expressed as percentage of the dry matter of films solubilized in distilled water.

#### 2.2.5. Mechanical Analysis

The mechanical properties of the films were measured using a SANTAM machine (STM-5, SANTAM, Karaj, Iran), according to the ASTM method D882 [25]. Before testing, the prepared films were equilibrated at 50% RH in a desiccator containing saturated magnesium nitrate solution at 25 °C for 24 h. The initial grip separation and crosshead speed was adjusted at 50 mm and 5 mm/min, respectively. Two mechanical parameters, including tensile strength (TS) and elongation at break (EB), were calculated from force–extension curves. TS (MPa) was determined by dividing the maximum load on the film before rupture by the cross-sectional area of the initial sample, and EB (%) was defined as the ratio of the final length at the point of specimen failure to the initial length of a sample and expressed as a percentage.

#### 2.2.6. Antibacterial Properties

The antibacterial impacts of the fabricated films were tested against two important foodborne pathogens, *Staphylococcus aureus* (ATCC 25923) and *Salmonella* Typhimurium (ATCC 14028) by the disc diffusion method [26]. In brief, the stock cultures were prepared overnight in brain heart infusion (BHI) agar (Merck, Darmstadt, Germany) at 37 °C. For the preparation of inoculums, the tested bacteria were transferred into BHI broth tubes and, after 18 h incubation at 35 °C, the bacterial density was adjusted to approximately 1 × 10^8^ CFU/mL using a spectrophotometer (Optizen 2120UV plus, Mecasys Co. Ltd., Daejeon, Korea) at 600 nm. Then, the CH films were cut into a 5 mm diameter circular disc and placed on the surface of pre-inoculated (with 100 μL of each bacterial suspension) Muller Hinton agar (MHA) Petri dishes and incubated at 37 °C for 24 h. Inhibition zones were then measured using a caliper and recorded.

#### 2.2.7. Fourier-Transform Infrared (FTIR) Spectroscopy

The interactions of CH and OG molecules in the structure of the developed composites were determined using a Bruker Equinox spectrometer (Bruker Banner Lane, Coventry, Bremen, Germany). In this regard, two samples, including pure CH and CH1/OG0.625 films, were selected and analyzed. Analyses were carried out in the range of 4000 to 400 cm^−1^ at a resolution of 4 cm^−1^. Sixteen scans were co-added for each spectrum.

#### 2.2.8. Film Morphology

The morphology of the films was studied by a scanning electron microscope (LEO-1450 VP, Zeiss, Oberkochen, Germany). Film’s samples were mounted on the specimen holder and sputter-coated with gold (10 mm) under vacuum, and SEM photographs were obtained.

### 2.3. Experimental Design and Statistical Analysis

To optimize the physical, mechanical and antibacterial properties of CH/OG composite, at the first step, a preliminary study was conducted using a full factorial design (FFD) to determine a primary proportion of CH and OG. It should be mentioned that the concentrations of CH and OG used for FFD were extracted from the relevant literature. The explanation of the detail of this step is unnecessary, and the data obtained by FFD was used for the next step.

Based on the FFD data, one proportion of CH/OG was selected (1:0.375 *w*/*w*). Then, in the following step, an RSM was run for assessing the transmittance, water barrier performance, water solubility, tensile strength (TS), elongation at break (EB) and antibacterial effects of the films. A central composite design (CCD), characterized by 13 experimental points (4 star points, 4 cube points, and 5 central points), was employed for fitting a second-order response surface. Table 1 represents the factors and their actual and coded values. The concentrations of CH and OG in the CCD were selected following the FFD data, and center point of the CCD was the established FFD CH/OG proportion (1:0.375 *w*/*w*).

Three replicate experiments were carried out at each design condition, and the mean values were recorded as observed responses. All experiments were performed in randomized order to minimize the impact of unexpected variability in the responses. Multiple regression analysis was used to examine the effects of CH (X_1_) and OG (X_2_) on responses (transmittance, WVP, water solubility, TS, EB and bacterial inhibition zone) by matching the obtained responses (*Y*) with the code factors using the polynomial function associated with the experimental condition (Equation (2)):Y = β_0_ + ∑ β_i_x_i_ + ∑ β_ii_x_i_^2^ + ∑ β_ij_x_i_x_j_(2)
where β_0_, β_i_, β_ii_ and β_ij_ are constant, first order linear, quadratic and interaction coefficients, respectively. X_i_, X_i_^2^, and X_ij_ represent linear, quadratic and interactive impacts of independent variables, respectively. The coefficients of the independent factors were estimated by multiple regression analysis and evaluated by analysis of variance (ANOVA). The Tukey test was applied at significance levels of 0.05 and 0.01 to determine differences among treatments. Design-Expert^®^ software (Version 7, Stat-Ease, Inc., Minneapolis, MN, USA) was used to perform regression analysis, the experimental design matrix, development of surface plots and the optimization procedure.

## 3. Results and Discussion

### 3.1. Statistical Analysis and Model Fitting

Response surface methodology (RSM) is a collection of mathematical, statistical and theoretical techniques used for model building to optimize the level of independent variables [27]. Table 2 presents the experimental design and results obtained for the physical, mechanical and antibacterial response variables. Polynomial equation coefficients were computed from experimental results to predict the response variables. In this regard, regression equations for each response, obtained from RSM are mentioned in the following equations (Equations (3)–(9)):WVP = 7.77101 − 10.0471 X_1_ − 10.434 X_2_ + 1.28 X_1_X_2_ + 4.6469 X_1_^2^ + 13.7876 X_2_^2^.(3)
Transmittance = 76.0376 + 4.78789 X_1_ + 12.0758 X_2_ − 11.9051 X_1_X_2_ − 1.7491 X_1_^2^ − 3.06138 X_2_^2^(4)
TS = 5.95927 + 14.7547 X_1_ + 24.3854 X_2_ − 7.92 X_1_X_2_ − 3.47069 X_1_^2^ − 12.3628 X_2_^2^(5)
EB = 28.0783 + 33.4023 X_1_ + 20.7701 X_2_ − 24 X_1_X_2_ + 1.96552 X_1_^2^ − 80.1379 X_2_^2^(6)
Inhibition zone of *S. aureus* = 7.49103 + 1.0075 X_1_ + 0.3825 X_2_ − 0.0575 X_1_X_2_ + 0.0644397 X_1_^2^ + 0.0444397 X_2_^2^(7)
Inhibition zone of *S.* Typhimurium = 3.86154 + 0.603678 X_1_ + 0.712184 X_2_ + 0.56 X_1_X_2_ + 2.72483 X_1_^2^ − 0.94069 X_2_^2^(8)
Water solubility = 50.89979 − 4.50667 X_1_ − 106.41667 X_2_ +10.56 X_1_X_2_ − 0.37 X_1_^2^ + 113.16 X_2_^2^(9)
where X_1_ and X_2_ represent CH and OG, respectively.

ANOVA and regression coefficients for the prediction models fitted to the experimental response values are summarized in Table 3 and Table 4, respectively. The quadratic model was found to sufficiently describe the experimental findings of the response variables, without a significant lack of fit (*p* > 0.05) and with the coefficient of determination (R2) values of 0.9531, 0.9911, 0.9127, 0.8689, 0.951, 0.9783 and 0.8766 for *S. aureus*, *S.* Typhimurium, transmittance, WVP, water solubility, TS and EB, respectively. In this study, closeness to unit R^2^ demonstrates that the effect of CH and OG on dependent variables could be sufficiently described by a quadratic polynomial model. ANOVA was used to determine the significance level of polynomial equation coefficients. Smaller *p*-value indicates a significant effect of any independent variable on the response variable.

### 3.2. FTIR Analysis

FTIR spectroscopy was used to better understand the molecular interactions of CH and OG in the CH/OG composites, and to evaluate the obtained physical and mechanical tests’ results. The spectra of CH and CH1/OG0.625 films are depicted in Figure 1. The broad bands at 3700–3000 cm^−1^, centered at 3386 and 3361 for CH and CH/OG films, respectively (Figure 1a,b), can be assigned to OH stretching modes that are associated with free, inter and intramolecular hydrogen bonds of the films’ components, including CH and OG, and the bonds in the region of 3000–2800 cm^−1^ are related to C–H stretching. The spectrum of the CH film showed a carbonyl stretching peak, a NH_2_ bending peak and a CH_2_ bending at 1641, 1558 and 1406 cm^−1^, respectively (Figure 1a). These bands were similarly observed in the spectra obtained for CH films by Antoniou et al., 2015 and Sakloetsakun et al., 2015 [1,11]. In fact, the presence of N-acetylglucosamine in the structure of CH was mentioned as the probable element creating the carbonyl stretching peak [11]. With the addition of OG, a shift in in the amino group (NH_2_) bending peak to a lower wavenumber (1529 cm^−1^) was observed (Figure 1b), suggesting an electrostatic interaction between the positively charged protonated amine groups of CH and the negatively charged groups of OG. Additionally, the CH_2_ bending and carbonyl stretching peaks were shifted to lower wavenumbers (Figure 1b), probably due to the formation of intermolecular hydrogen bonds between CH and OG. These observations and changes were previously reported for the combination of CH with various gums showing the importance of electrostatic interactions and hydrogen bonds in the structure of CH/gum composites [11,13].

### 3.3. Film Microstructure

SEM micrographs of the cross-section of CH and CH/OG films are shown in Figure 2. The CH film was smooth and transparent and showed a yellowish tinge (Figure 2 and Figure 3), while the addition of OG to CH film resulted in less transparent and coarser films with a darker color appearance. In fact, the coarser microstructure of CH/OG film can be explained by the fact that CH and OG chains interact by the formation of electrostatic and intermolecular hydrogen bonds and entanglement, which might, to some extent, have an effect on the inter/intramolecular forces acting on CH [11]. FTIR findings also support this idea. Xu et al., 2019 and Sakloetsakun et al., 2016 similarly observed coarser structure of CH/gum films in SEM images for higher ratios of gum arabic/CH (4:1 and 1:1) composites [11,12].

### 3.4. DSC Analysis

The impact of OG on the thermal properties of CH-based films was studied by DSC examination. The DSC thermograms of neat CH and CH1/OG0.625 films are shown in Figure 4. The evaluation of DSC thermogram showed two important endothermic peaks for both films: first, between 84–96 °C for the CH film and between 60–69 °C for CH1/OG0.625; and second, between 214–222 °C for the CH film and between 220–226 °C for the CH1/OG0.625 composite.

The recorded endothermic peaks, also called dehydration temperature, can be assigned to the loss of water, while the second endothermic ones can be ascribed to the melting temperatures (T_m_) of the films [28]. As can be seen in Figure 4, the addition of OG to the CH film resulted in a lower dehydration temperature compared to the CH film. This is not in agreement with the DSC data reported by Sakloetsakun et al., 2016, who mentioned that the incorporation of different percentages of gum arabic to CH films did not significantly alter the dehydration temperatures of the resultant composites [11]. The melting temperature of the CH film containing OG was higher compared to the neat CH film. This could be related to the interactions between the CH and OG chains, as shown by FTIR results. In this regard, de Morais Lima et al., 2017, who observed a similar phenomenon for CH films containing xanthan gum, mentioned that the higher T_m_ is probably related to the lower mobility of biopolymers chains associated with inter-polyelectrolytes electrostatic interactions [13].

### 3.5. Transmittance

The light transmittance of a food packaging film affects the rate of oxidation of lipids, particularly in high-fat food products, such as different types of meats. On the other hand, highly transparent films might be more appealing for the presentation of a food product in the market. Therefore, based on the application of a packaging film, different levels of transparency might be desirable. Although the light of short wavelengths (<400 nm, like UV light) has the greatest impact on the oxidation of lipids in foods, longer wavelengths also have the ability to cause changes in foodstuffs [29]. In the present study, the transmittance of light in the wavelength of 660 nm was evaluated, and a higher rate of transmittance was more desirable for the studied composites. The visual appearances of selected CH/OG composites are shown in Figure 3.

The transmittance of the films was mainly depended on the concentration of CH (Table 3) as it had a significant linear effect on the transmittance (*p* < 0.01), followed by the main effect of OG (*p* < 0.05) and the negative regression coefficients for CH (−0.794), and OG (−0.266) indicated that the transparency of the film decreased when CH or OG content increased. On the other hand, the quadratic impacts of CH and OG and the interaction between CH and OG were not significant (*p* > 0.05).

The relationship between the two variables and experimental findings on transmittance can be better comprehended by examining the surface plot shown in Figure 5a, in which the impact of the independent variables (CH and OG) on the transparency of the composites were evaluated. Figure 5a shows that transmittance of the composites linearly decreased with the increase in CH and OG concentrations, and the linear negative impact of CH was more apparent at higher concentrations of OG.

The effects of the addition of different gums on the transparency of various films, including chitosan, were investigated. In a study by Rao et al., 2010, chitosan films were produced by the casting method using different ratios (15–50% *v*/*v*) of guar gum [14]. The addition of the gum changed the transparency of films, and a higher percentage of guar gum resulted in increased transparency. They also reported that the interaction with the water molecules and gum modified the refractive index of chitosan, consequently affecting the composite transparency. These findings are in agreement with the results of Xu et al., 2019, who showed that higher ratios of gum arabic decreased the opacity of chitosan film [12]. However, the distinct relationship between the concentration of gum and transparency was not observed in the present study. This could be due to the different interaction of OG with CH, and the existence of some light barrier substances in the tested OG. Nur Hazirah et al., 2016 also reported that increasing the concentration of xanthan gum from 5 to 25% (*w*/*w* solid) in the blended gelatin-carboxymethyl cellulose films significantly (*p* < 0.05) decreased transparency [30]. They suggested darker colors caused by crosslinking of xanthan gum with gelatin-CMC as the probable source of lower transparency, and this may justify our finding.

The impact of the concentration of chitosan on the light transmission rate was also previously evaluated. For instance, the optimization of chitosan levels revealed that higher levels of chitosan lessen the degree of transparency of resultant films [21]. This is in line with our result in which a negative coefficient was established for the chitosan variable. In fact, it is not surprising that higher levels of a film-forming polymer can lower the degree of transparency.

### 3.6. Water Vapor Permeability

The moisture barrier capability of food packaging material plays a pivotal role in preserving the quality of fresh foodstuffs. In general, polysaccharide films have higher WVP in comparison with petroleum-based plastics. Therefore, improving the water vapor barrier performance of these materials can accelerate their commercialization process. The WVP of the developed composites ranged from 0.6 to 2.12 (10^−8^ g/m h Pa), which are comparable with the values that exist in the literature [31,32]. Based on the ANOVA data (Table 3), the quadratic terms of CH and OG were significant (*p* < 0.01), while the main interaction effects were not (*p* > 0.05). Both quadratic regression coefficients of CH and OG were positive, showing that higher values of the independent variables resulted in higher water vapor permeability. Figure 5b represents the variation of WVP of a chitosan-based film as a function of CH and OG concentration. As can be seen, both CH and OG exert a quadratic effect on the WVP of the films. Within the tested concentrations of the film constituents, middle CH and OG contents led to better barrier performances. Although studies are lacking in addressing the effects of OG on barrier performance of food packaging materials, the impacts of other gums on water permeability of various bio-based polymers, such as chitosan, were previously evaluated. For example, contrary to our findings, it was shown that varying proportions of xanthan gum (10–50% *w*/*w*) or arabic gum (20–50% *w*/*w*) had no significant effect on water vapor permeability of low and high molecular weight chitosan-based films [11,13].

However, positive impacts of gums on water barrier performance of chitosan films were also observed, where the addition of different ratios of arabic gum (0.2–0.8 *w*/*w*) into the matrix of chitosan films decreased the WVP of the resultant composites [12]. It should be mentioned that this decrease was not dependent on the proportion of gum. In fact, the addition of 0.33 *w*/*w* of gum was sufficient to bring down the WVP to the minimum value compared to pure chitosan film (*p* < 0.05). The author attributed this finding to the weakening of hydrogen binding between CH and water molecules and the compact structure of the chitosan–gum film. In this regard, here in this study, electrostatic interactions between CH and OG compounds (at the middle concentrations) may lead to the formation of a more compact and consequently stronger barrier structure that limits the permeation of water vapor through the film. Nevertheless, the exact mechanism of those changes in permeability is still unknown.

### 3.7. Water Solubility

The water solubility of the CH-based composites ranged from 24.2 to 35.5% (Table 2). ANOVA data showed that OG could linearly and quadratically affect the solubility of the polymers (*p* < 0.05), but the interaction impact of OG and CH, and the CH effect, were not significant. The negative linear term (−0.32) and positive quadratic coefficient (1.77) of OG showed that the incorporation of lower concentrations of this material led to a lower percentage of water solubility, while a higher OG percentages increased the water solubility (Table 4). The effects of CH and OG on the water solubility of the films are well illustrated in Figure 5c, where the minimum values were recorded for the middle concentrations of OG.

The water solubility of food packaging film is among the most important characteristics of biopolymer that may limit the application of those materials, especially in fresh food products. From the food application point of view, polymers with lower levels of water solubility are generally desirable. In this study, the linear effect of OG on solubility parameters was encouraging. Rahman et al., 2021 also reported that the solubility of CH/guar gum composite (15.26%) was ~50% lower than the value of neat CH film (29.46%) [33]. The addition of CH to tara gum film similarly significantly decreased the water solubility of the composites (*p* < 0.05) [1]. It seems that the electrostatic interactions and the strong hydrogen bonding between CH and gum are the main factors that brought down the solubility of the resultant polymers [33,34].

### 3.8. Mechanical Properties

Concerning mechanical characteristics, an edible film should protect the food products against shocks and stresses, and maintain the integrity of the package during transportation and distribution. TS and EB are among the key mechanical parameters that are usually evaluated for the characterization of various packaging polymers. TS represents the maximum tensile stress that a polymer can tolerate, and EB characterizes the highest variation of a film before breakage [35]. The mechanical properties of polymers can be affected by several factors, especially by the interactions between their constituents.

Based on TS results (Table 3), both linear and quadratic impacts of CH and OG were significant (*p* < 0.05), and the linear effects of the compounds were more pronounced (*p* < 0.01).

This can be better understood by examining the surface plot illustrated in Figure 6a, in which the TS values were linearly increased with the increase in CH and OG contents of the films. As it can be seen in Figure 6a, the rate of TS increase is declined with the higher percentages of both independent variables.

This phenomenon was previously stated by several authors. For instance, Rao et al., 2010 and Xu et al., 2019 reported that the chitosan films contained lower concentrations of the added gums (15% *v*/*v* of guar gum and 20% *w*/*w* of arabic gum) showed the highest TS, while further increase in the gums content in the blend resulted in lower tensile strength values (*p* < 0.05). The high TS values of those CH films were connected to the formation of intermolecular hydrogen bonds between OH^-^ of gum and NH_3_^+^ of the CH backbone, and the evidence of the presence of those bonds were demonstrated in the FTIR results of the present study. Furthermore, the more entangled microstructure of the films containing higher percentages of gum was mentioned as the probable cause of TS decrease. The linear and quadratic terms of OG in the present study agree with the aforementioned findings.

According to the ANOVA results for the fitting model of EB experimental data (Table 3), the main effects of both CH and OG were significant (*p* < 0.01), while the independent variables could not quadratically change the response (*p* < 0.05). The regression coefficients for the fitted model show that EB values increased linearly with the increase and decrease in CH and OG, respectively (Table 4). These effects are well illustrated in Figure 6b, where the maximum EB value was recorded at the point of maximum concentration of CH (1.5% *w*/*v*) and minimum OG percentage (0.125% *w*/*v*).

The reduction in the extensibility of chitosan films incorporated with gums was not a new experience. In line with our results, different studies also show that, with the addition of different gums, including tara gum [1] and arabic gum [11,12], more brittle chitosan films were yielded. Antoniou et al., 2015 produced and characterized tara gum edible films incorporated with different ratios (0–15% *w*/*w*) of chitosan and chitosan nanoparticles. They reported that the EBs of the films decreased slightly when CH was added, and EB % was significantly lower than the control, only when the concentration was 15% *w*/*w* (*p* < 0.05). Sakloetsakun et al., 2016 also reported the lower percentages of EB for chitosan polyelectrolyte complex films with 50 and 37.5% *w*/*w* arabic gum [11]. Several reasons have been proposed for those effects, such as weakening hydrogen binding, decreased plasticizing effects, the globe-like microstructures and higher strength of the developed films [1,11]. It seems that the polysaccharides fraction of OG acted like the aforementioned gums in bringing down the extensibility of the developed CH/OG composites.

Different studies were carried out regarding the influence of CH concentration on mechanical properties. Besides chitosan, various factors such as glycerol concentration and nanofiller compound were examined in those studies [4,36]. The range of studied chitosan concentrations was close to the CH concentrations tested in the present work. However, the impact of CH on TS and EB differs among those studies and our finding, showing the importance of the second material combined with CH.

### 3.9. Antibacterial Properties

Although the primary objective of the incorporation of OG into CH film was to enhance the physical and mechanical properties, the evaluation of the antimicrobial activity of the composites was also worthy, especially due to the well-known antimicrobial effects of CH and probably frankincense. Among the various types of antibacterial testing used for the evaluation of active packaging polymers, a disc diffusion assay is widely employed. In fact, this method can simply simulate wrapping foodstuffs and be used as a quantitative examination.

In this regard, all of the fabricated films could create some clear inhibition zones against the growth of both *S. aureus* and *S.* Typhimurium (Table 2).

Based on ANOVA data (Table 3), the main effects of CH against both strains were significant (*p* < 0.01), while OG could linearly increase the antibacterial effects of the active films only against *S. aureus* (*p* < 0.05). Moreover, higher percentages of CH could significantly inhibit the growth of *S.* Typhimurium (*p* < 0.05).

Figure 7a,b depicts the effect of independent variables on the antibacterial responses. As it can be seen, the pattern of the inhibition zone changes is almost similar for both pathogens, and maximum bactericidal–bacteriostatic effects can be found at the point of maximum concentration of CH and OG. Moreover, the linear antibacterial effects of the independent variables are evident. A positive interaction between the lower concentration of CH and higher values of OG can readily be seen in *S. aureus’s* figure comparing to *S.* Typhimurium.

The antibacterial effects of CH-based film are extensively reported against many types of bacteria, including Gram-negative and Gram-positive food-borne pathogens [7]. Although the exact mechanism of CH antimicrobial effects is not recognized so far, the presence of cationic groups, such as protonated NH_3_^+^, in the chemical structure of the polymer is repeatedly proposed as a crucial factor. In fact, it is suggested that these positively charged groups interact with the electronegative components in bacterial cell membrane leading to the leakage of nutritious materials [37]. Moreover, chitosan molecules can probably participate on the bacterial cell surface and create an impervious barrier, disrupting the transport of vital nutrients into the cell and therefore resulting in bacterial death [38]. Therefore, the significant antibacterial impacts of CH can be explained by the aforementioned mechanisms.

The antimicrobial activity of the essential oil of OG (or frankincense) obtained from different species of *Boswellia* has been frequently reported [19,39]. When this active constituent was added to different films (i.e., alginate and carboxymethyl cellulose/chitosan), it could successfully enhance the antimicrobial properties of the resultant composites [40,41]. Moreover, the antibacterial effects of boswellic acids were also demonstrated against different pathogenic bacteria [42].

In this study, during the preparation of CH/OG blends, a percentage of both boswellic acids and EO parts of OG were probably remained in the blends and acted against the tested bacteria. Additionally, a part of the significant linear antibacterial effect of OG against *S. aureus* could be attributed to the fact that EO is more effective against Gram-positive bacteria [43].

Similarly, the combination of CH and gums has previously led to the formation of effective antimicrobial films. Rao et al., 2010 reported that CH/guar gum films were active against *S. aureus* and *Escherichia coli*, and the addition of the gum at 15% (*v*/*v*) resulted in additional antibacterial activity against *E. coli*. However, the incorporation of higher percentages of guar gum led to a decrease in antibacterial activity. The formation of intermolecular hydrogen bonding between NH_3_^+^ of CH and the hydroxyl group of the gum was mentioned as the probable cause of the lower activity. In fact, this phenomenon might be the reason why the quadratic term of OG was negative (in the case of *S.* Typhimurium) and lower than the relevant linear effects for both bacteria.

### 3.10. Multiple Responses Optimization

Based on the results obtained from each response surface model, the overall desirability function was used to obtain an optimum antibacterial film with satisfactory physical, mechanical and antibacterial properties for food application. For this purpose, responses such as transparency, TS, EB and bacterial inhibition zones were desired to be at their maximum values, while lower values of WVP and water solubility were targeted. Considering all variables and responses, the optimum condition for CH/OG film with the most desirable properties (with the composite desirability of 60%) was at 1.31% (*w*/*v*) CH and 0.3% (*w*/*v*) OG concentrations (Table 5).

## 4. Conclusions

In the present study, we evaluated the physical, mechanical and antibacterial properties of newly fabricated CH/OG blend films by response surface methodology. Based on the physical evaluation, the transparency of the films was mainly decreased by the CH factor followed by OG, and the developed films showed the highest water vapor barrier performance and the lowest water solubility when the middle concentrations of CH (near 1% *w*/*v*) and OG (near 0.375% *w*/*v*) were used. Mechanical examination revealed that, while CH could positively enhance TS and EB values, only the tensile strength of the films was increased by OG, and the higher percentages of the gum resulted in lower EB percentages. Finally, antibacterial testing showed that both independent variables had antibacterial effects, and CH was more efficient against both tested Gram-positive and Gram-negative strains. The FTIR, thermal and SEM evaluations indicated that important interactions are created between the CH and OG chains. We found that 1.31% (*w*/*v*) CH and 0.3% (*w*/*v*) OG led to the highest desirability (60%) for final packaging films. Based on the positive data obtained from antibacterial, physical and mechanical analyses, it can be concluded that the fabricated composite can be used in active food packaging, especially for preserving perishable meat and horticultural products.

## Figures and Tables

**Figure 1 polymers-14-03960-f001:**
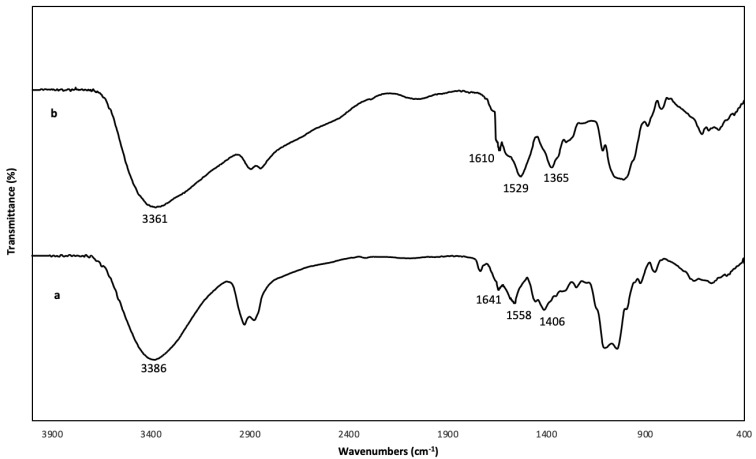
FTIR spectrum of pure chitosan (CH) film and CH film incorporated with 0.625% *w*/*v* olibanum gum (OG): (**a**) (pure CH); (**b**) (CH1/OG0.625).

**Figure 2 polymers-14-03960-f002:**
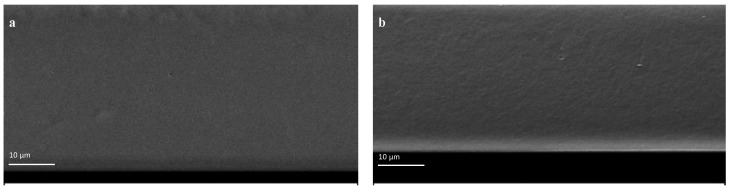
SEM photographs of the cross-section of pure chitosan (CH) film and CH film incorporated with 0.625 %*w*/*v* olibanum gum (OG): (**a**) (pure CH); (**b**) (CH1/OG0.625).

**Figure 3 polymers-14-03960-f003:**
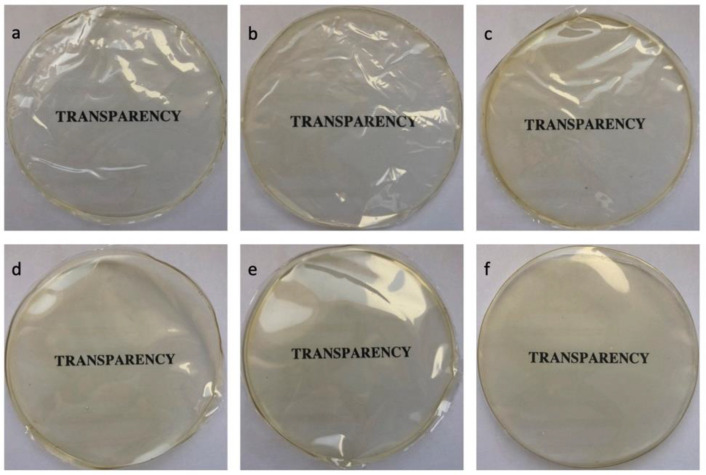
Visual aspects of the chitosan (CH) films incorporated with different concentrations of olibanum gum (OG): (**a**) (CH0.5/OG0.375); (**b**) (CH0.75/OG0.25); (**c**) (CH1/OG0.125); (**d**) (CH1/OG0.625); (**e**) (CH1.25/OG0.5); (**f**) (CH1.5/OG0.375).

**Figure 4 polymers-14-03960-f004:**
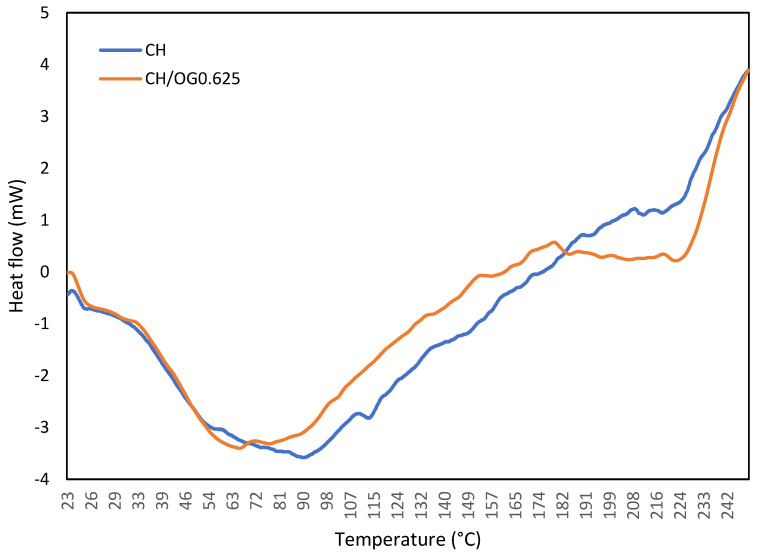
DSC thermograms of pure chitosan (CH) film and CH film incorpo-rated with 0.625 %*w*/*v* olibanum gum (OG): (blue line) (pure CH); (orange line) (CH1/OG0.625).

**Figure 5 polymers-14-03960-f005:**
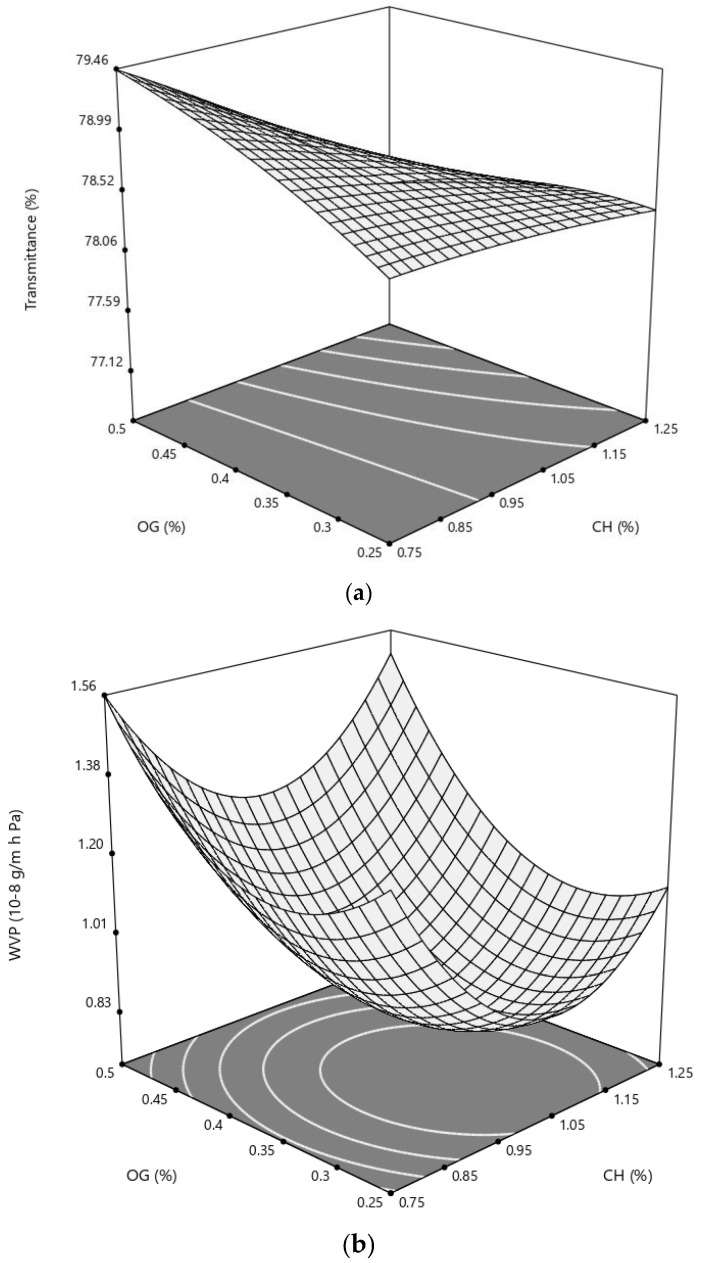
Response surface of (**a**) transmittance, (**b**) water vapor permeability (WVP) and (**c**) water solubility of a chitosan-based film as a function of chitosan (CH) and olibanum gum (OG) (% *w*/*v*).

**Figure 6 polymers-14-03960-f006:**
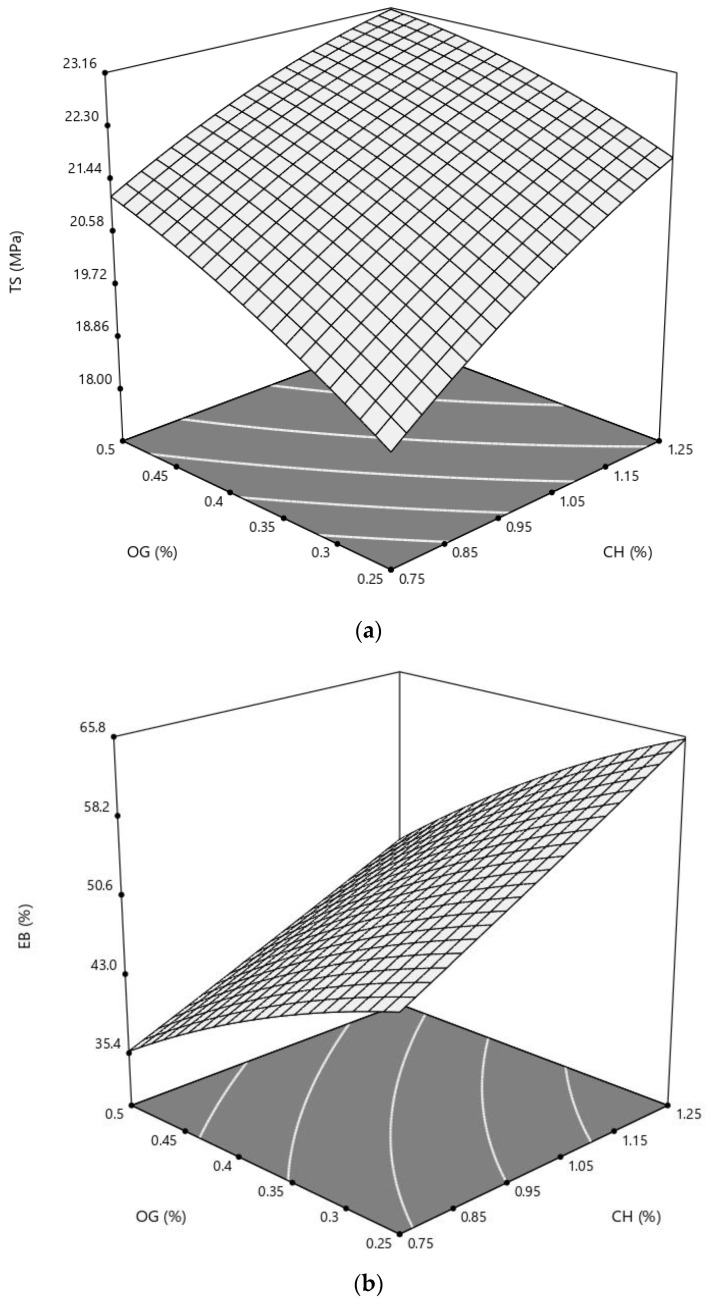
Response surface of (**a**) tensile strength (TS) and (**b**) elongation at break (EB) of a chitosan-based film as a function of chitosan (CH) and olibanum gum (OG) (% *w*/*v*).

**Figure 7 polymers-14-03960-f007:**
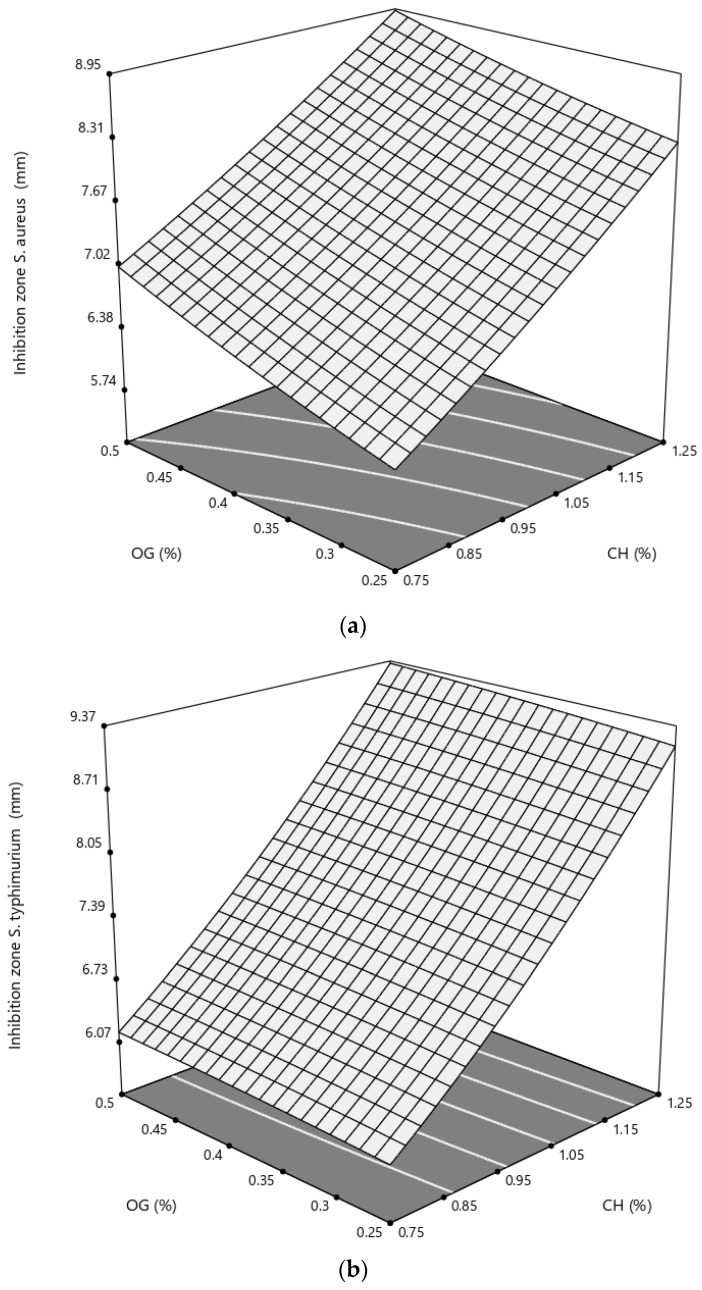
Response surface of antimicrobial activity of a chitosan-based film against (**a**) *S. aureus* and (**b**) *S.* Typhimurium as a function of chitosan (CH) and olibanum gum (OG) (% *w*/*v*).

**Table 1 polymers-14-03960-t001:** Independent variables and their coded and actual values used in the central composite design.

Independent Variables	Symbol	Levels				
		−2	−1	0	1	2
Chitosan (% *w*/*v*)	CH	0.5	0.75	1	1.25	1.5
Olibanum gum (% *w*/*v*)	OG	0.125	0.25	0.375	0.5	0.625

**Table 2 polymers-14-03960-t002:** Central composite design with experimental values of response variables.

	Variable Levels		Responses						
Run	CH ^a^ (% *w*/*v*)	OG ^b^ (% *w*/*v*)	Inhibition Zone (mm)		Transmittance (%)	WVP ^c^ (10^−8^ g/m h Pa)	Water Solubility (%)	TS ^d^ (MPa)	EB ^e^ (mm)
			*S. aureus*	*S.* Typhimurium					
1	0.75	0.25	6.12	5.82	78.87	1.66	30	19	46
2	1.25	0.25	8.22	9.25	78.15	1.45	29.16	21.5	60
3	0.75	0.5	7.01	5.9	79.2	1.55	26.3	21.5	35
4	1.25	0.5	8.88	9.4	77	1.5	26.78	23.01	46
5	0.5	0.375	5.74	5.3	80.05	2.1	26.61	18.19	38
6	1.5	0.375	9.8	11.23	76.75	1.82	24.86	23.45	68
7	1	0.125	6.93	7.37	79.24	1.2	35.5	19.22	65
8	1	0.625	8.45	7.68	78.05	2.12	30.3	22.61	30
9	1	0.375	7.5	7.57	78.6	0.6	25.92	21.52	45
10	1	0.375	7.8	7.6	79.42	0.7	24.2	21.44	50
11	1	0.375	7.2	7.7	78.7	1	26.91	22.16	60
12	1	0.375	7.02	7.4	78.59	0.9	26.47	21.61	54
13	1	0.375	8.02	7.6	78.74	0.87	25.8	21.7	52

^a^ CH, Chitosan. ^b^ OG, Olibanum gum. ^c^ WVP, Water vapor permeability. ^d^ TS, tensile strength. ^e^ EB, elongation at break.

**Table 3 polymers-14-03960-t003:** Analysis of variance (ANOVA) for the prediction models fitted to experimental response values obtained from the optimization of the concentration of chitosan (CH) and olibanum gum (OG) into film forming solution (*p*-Value) ^a^.

Source	Inhibition Zone (mm)		Transmittance (%)	WVP ^b^ (10^−8^ g/m h Pa)	Water Solubility (%)	TS ^c^ (MPa)	EB ^d^ (mm)
	*S. aureus*	*S.* Typhimurium					
X_1_ (CH)	<0.0001 **	<0.0001 **	<0.0001 **	0.359	0.2412	<0.0001 **	0.002 **
X_2_ (OG)	0.004 **	0.252	0.036 *	0.071	0.0009 **	<0.0001 **	0.001 **
X_1_^2^	0.359	0.004**	0.184	0.001 **	0.9024	0.011 *	0.915
X_2_^2^	0.521	0.731	0.54	0.004 **	<0.0001 **	0.019 *	0.296
X_1_ X_2_	0.725	0.864	0.075	0.75	0.473	0.147	0.785
Regression	<0.0001 **	<0.0001 **	0.001 **	0.005 **	0.0002 **	<0.0001 **	0.004 **
*R* ^2^	0.9531	0.9911	0.9127	0.8689	0.951	0.9783	0.8766
*R*^2^ (adjust)	0.9195	0.9848	0.85	0.7752	0.9159	0.9628	0.7884
*R*^2^ (pred)	0.9268	0.9293	0.6442	0.1119	0.8617	0.8727	0.5203
Lack of fit	0.994	0.054	0.442	0.111	0.8023	0.374	0.539

^a^ *, significant at *p* < 0.05; **, significant at *p* < 0.01. ^b^ WVP, Water vapor permeability. ^c^ TS, tensile strength. ^d^ EB, elongation at break.

**Table 4 polymers-14-03960-t004:** Regression coefficients (coded) for the fitted models.

Source ^a^	Coefficients						
	Inhibition Zone (mm)		Transmittance (%)	WVP ^c^ (10^−8^ g/m h Pa)	Water Solubility (%)	TS ^d^ (Mpa)	EB ^e^ (mm)
	*S. aureus*	*S.* Typhimurium					
Constant	7.49	7.53	78.71	0.877	25.99	21.679	50.97
X_1_ (CH ^a^)	1	1.56	−0.794	−0.0683	−0.3217	1.2108	7.08
X_2_ (OG ^b^)	0.38	0.07	−0.266	0.1483	−1.37	0.8992	−7.92
X_1_^2^	0.06	0.17	−0.1093	0.2904	−0.0231	−0.2169	0.12
X_2_^2^	0.04	−0.01	−0.0478	0.2154	1.77	−0.1932	−1.25
X_1_ X_2_	−0.05	0.01	−0.372	0.04	0.33	−0.247	−0.75

^a^ CH, Chitosan. ^b^ OG, Olibanum gum. ^c^ WVP, Water vapor permeability. ^d^ TS, tensile strength. ^e^ EB, elongation at break.

**Table 5 polymers-14-03960-t005:** Optimization of CH/OG ^a^ film preparation (at desirability values of 1.31% *w*/*w* chitosan and 0.3% *w*/*w* olibanum gum).

	Factor						
	Inhibition Zone (mm)		Transmittance (%)	WVP ^b^ (10^−8^ g/m h Pa)	Water Solubility (%)	TS ^c^ (MPa)	EB ^d^ (mm)
	*S. aureus*	*S.* Typhimurium					
Goal	8.66	9.67	77.98	1.19	26.82	22.4	64.89
Optimized value	Maximize	Maximize	Maximize	Minimize	Minimize	Maximize	Maximize

^a^ CH/OG, chitosan/olibanum gum. ^b^ WVP, Water vapor permeability. ^c^ TS, tensile strength. ^d^ EB, elongation at break.

## Data Availability

Not applicable.

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
