# Peer review of "Optimization of Antibacterial, Physical and Mechanical Properties of Novel Chitosan/Olibanum Gum Film for Food Packaging Application"

_polymers, 2022, doi:10.3390/polym14193960_

Round 1

Reviewer 1 Report

The paper "Optimization of the antibacterial, physical and mechanical properties of the new chitosan/olibanum gum film for application in food packaging" presents a representative analysis of the proposed study, and I recommend publication after the following corrections:

Minor revision:

#Line103: The unit must be separated from the value.

Major revision:

The work presents a good structure and a good literature review that corroborates the results obtained. However, as it is a promising material for use as food packaging (as mentioned by the authors), I strongly suggest that the authors do more characterizations such as thermal stability, solubility and if possible morphology.

Reviewer 2 Report

I have the following comments:

  1. There is no clearly defined aim in the abstract.
  2. Line 47: there is no reference.
  3. Line 56: there is no reference.
  4. Line 74: words FOR THE FIRST TIME should be erases, there are not appropriate.
  5. The following reference should be used: Jancikova, S., Dordevic, D., Tesikova, K., Antonic, B., & Tremlova, B. (2021). Active edible films fortified with natural extracts: Case study with fresh-cut apple pieces. Membranes, 11(9), 684.
  6. Why authors do not have exact numbers obtained by analysis, but only RSD. It seams that authors published already data? If that is correct it should be provided reference of it, and then reviewers can consider if only statistical analysis can be published separately. Authors should explain it.

Round 2

Reviewer 2 Report

The manuscript can be accepted.